# Estimating the Economic Values of Restricted Monoculture *Eucalyptus* Plantations: A Choice Modeling Approach

**DOI:** 10.3390/ijerph19159073

**Published:** 2022-07-26

**Authors:** Amare Tesfaw, Feyera Senbeta, Dawit Alemu, Ermias Teferi

**Affiliations:** 1Department of Agricultural Economics, College of Agriculture and Natural Resources, Debre Markos University, Debre Markos P.O. Box 269, Ethiopia; 2Center for Environment and Development Studies, Addis Ababa University, Addis Ababa P.O. Box 1176, Ethiopia; feyera.senbeta@aau.edu.et (F.S.); ermias.teferi@aau.edu.et (E.T.); 3Bilateral Ethio-Netherlands Effort for Food, Income & Trade Partnership (BENEFIT), Addis Ababa P.O. Box 88, Ethiopia; dawit96@yahoo.com

**Keywords:** choice experiment, *Eucalyptus*, monoculture plantations, non-market benefit, marginal willingness to pay

## Abstract

Today, evaluating ecological wellbeing and ecosystem services is becoming a great concern towards conserving the natural resource base. Healthy functioning ecosystems have fundamental roles for aiding humankind to lead a healthy life and ensure an improved social welfare. Estimating the non-market benefits of ecosystem services can help experts and the public frame policy directions designed for landscape development. The ecosystem of the *Eucalyptus* hotspot highlands of northwestern Ethiopia, where this study was carried out, provides services that are essential to changes in the life of the society and biodiversity. However, in recent years, the ecosystem is facing a serious threat from intensive monoculture plantations of *Eucalyptus*. This has resulted in transformation of the cultural landscapes and a loss of biodiversity. The problem in turn calls for designing appropriate ecological improvement programs. Thus, the current study examined the preferences of residents concerning this area and estimated their willingness to pay (WTP) for the proposed ecosystem improvement programs using a Choice Experiment approach. Data were aggregated from 388 residents using a questionnaire survey in January 2020. The survey contained ecological improvement schemes and a hypothetical event by which respondents expressed their willingness to pay a yearly utility fee as a compensation for the improvement programs. Results showed significant differences in resident preferences towards the proposed ecological improvement attributes. The findings also indicated that the socioeconomic backgrounds of residents contributed for the heterogeneity in their WTP for ecological improvement schemes. Accordingly, the marginal willingness to pay of residents was USD 205/person/year for the respective ecological improvement attributes. The findings suggest that policy makers should consider such attribute-based public preferences while planning landscape development and conservation programs. This study can provide vital policy implications and contribute to knowledge as it presents how the non-market valuations of ecosystems help maximize social welfare.

## 1. Introduction

The northwestern highlands of Ethiopia have a dynamic ecosystem, which experiences intensive farming practices involving cultivation of food crops and plantations of *Eucalyptus* that sustain the livelihood of smallholder growers. Increasing population together with intensive traditional farming has resulted in degradation of the natural environment. The decline in productivity of crop lands has aggravated the rapid conversion of crop lands and pushed farmers to shift to plantation growing of the most rewarding tree genera, *Eucalyptus*. Today, *Eucalyptus* is among the most economically important for short rotation plantations as an option tree for biomass and fiber production [1,2]. *Eucalyptus* was introduced to Ethiopia in the second half of the 19th century for the rising demands of forest products for construction and fuel [3].

There are different reasons for the expansion of *Eucalyptus* in Ethiopia. The first and most prominent reason is the absence of native tree species that can be grown alternatively with equivalent economic contributions similar to *Eucalyptus*. Indigenous and other exotic forest stands, such as *Cordia africana*, *Juniperus procera*, *Podocarpus falcatus*, *Olea europaea*, *Acacia dicurens*, and *Cupressus* species, have limited growth and production capacity compared to *Eucalyptus* [4]. Another positive attribute of the tree that drive the expansion of *Eucalyptus* use, is its vigorous growing habit, high potential for coppicing, high yielding potential in short rotations, and the provision of fiber and energy sources suitable for a variety of uses [5,6,7,8]. The tree exhibits important traits, such as a lesser labor requirement [9,10] and higher water use efficiency [11,12,13], that help make it an excellent biofuel crop [14].

Despite its economic and social benefits, there is a growing concern regarding the negative ecological attributes and invasive nature of the tree, which can be considered as one of the greatest threats to biodiversity and the provision of valuable ecosystem services [15]. Different studies have criticized the tree for its ecological challenges, including for its invasive behavior [16], impact on soil health, depletion of ground water, depletion of nutrients, suppression of undergrowth and subsequent reduction in species in neighboring crops [17]. *Eucalyptus* also depletes ground water, which aggravates watershed degradation [18]. Different studies have evidenced that the same traits that make *Eucalyptus* a highly productive crop may contribute to its potential to be an invasive species [19]. *Eucalyptus globulus* is the most widely grown and well-adapted non-native tree species in the study area. Non-native species have the potential to influence the composition, structure, and function of ecosystems where they are introduced [20,21], and numerous reports suggest that species of *Eucalyptus* are capable of invading ecosystems across the world [22,23]. *Eucalyptus globulus* has been introduced and become established in five different countries with climates ranging from shrub lands to forests and resulted in the understory exhibiting decreased height and species richness [24].

Aside from the economic and social benefits, there are scientific debates regarding the negative ecological impacts *of Eucalyptus* and its expansion as a monoculture plantation. The introduction of *Eucalyptus*, as an alien genus to a new area, has serious invasion risks on biodiversity and ecosystem. Forstmaier et al. [16] reported the negative ecological impact of *Eucalyptus* and its invasive behavior on the ecological wellbeing. Various studies have also reported the negative impact of *Eucalyptus* on soil and biodiversity, with evidence that *Eucalyptus* can drain water resources, aggravate soil erosion, suppress undergrowth, deplete soil nutrients, and induce allelopathic effects [25]. Studies [2,17,18] have reported that *Eucalyptus* has allelopathic effects around the root zones of the neighboring crops and ecological impacts due to its shading effect under and near the tree canopy. A study by Lopez [26] has reported a change in native Californian ecosystem processes following the invasion of the local landscapes by *Eucalyptus globulus*, which resulted in reduction in biological diversity due to the displacement of native plants and wildlife habitats. Teketay [3] has also reported a similarly conclusion that showed a lower herbaceous species richness in *Eucalyptus* plantations than in indigenous forest stands.

Given the positive economic attributes and negative ecological impacts of the tree, the natural forests in these areas have been eliminated and replaced with a monoculture of *Eucalyptus* plantations exposing ecosystem vulnerability and challenges to the long-term ecological sustainability. Ongoing land degradation, climate change, ocean acidification, and land cover changes are reducing the quantity and quality of ecosystem services being provided globally [27]. A study by Ali et al. [28] showed the impact of climate change on many organisms in different ecosystems. He found that annual mean precipitation was a key determinant of suitable habitat for ungulate species. Townhill et al. [29] also indicated that pressure on the Arctic species and the ecosystem was a result of human activities.

Practical evidence shows that the ecosystem of the *Eucalyptus* hotspot highlands of northwestern Ethiopia has been disturbed. This includes a near to complete replacement of the natural forests, conversion of crop lands to *Eucalyptus* plantations, and the establishment of a monoculture *Eucalyptus* plantation over most of the area. This apparently threatens the existing natural ecosystem that has provisioning, regulating, cultural, and supporting services [30].

To restore the natural ecosystem in the study area with Pareto efficiency land resource management, designing improvement programs for the restricted plantation of *Eucalyptus* would be of paramount importance. Natural resource management interventions by rational land use strategies are very important to control the rapid expansion of *Eucalyptus*, which can pose burdens on ecological wellbeing. In addition, it would be very important to estimate the economic value of restricted plantation areas of *Eucalyptus* and residents’ willingness to pay for the designed ecological improvement programs in the study area. It is possible to estimate the non-market benefits of restricting the expansion of *Eucalyptus* in economic terms and to examine the possible directions for supporting the conservation of ecological biodiversity that are preferred by people in the conservation strategy.

We employed a Choice Experiment (CE) method, which was selected as the most suitable technique to analyze people’s choice for its wide application for valuing environmental goods and services. The CE method is appropriate for analyzing choice to exploit the *Eucalyptus* hotspot highlands of the Northwestern Ethiopia and identify values on this basis.

The method has already been successfully applied in many different wetland settings and has the advantage of being able to generate multiple value estimates from a single application. Many scholars [31,32,33] have applied a CE for analyzing their studies. A CE has been widely used for analyzing people’s choice regarding wetland ecosystem improvements and providing policy suggestions including studies in USA and Canada [34], in Australia and Tasmania [35] and in Vietnam [36]. Other studies, like Luisetti et al. [37] used the same approach to examine readjustment of coastal policies in England.

The study hypothesized that the expansion of areas of monoculture *Eucalyptus* plantations in the *Eucalyptus* hotspot highlands of northwestern Ethiopia would impact the ecological wellbeing and that these ecological challenges can be overcome through designing appropriate ecological management programs. Hence, the main aim of this study was to contribute to these ecological improvement programs through estimating the economic value of restricted plantation areas of *Eucalyptus* in the study region. In addition, the study was intended to estimate residents’ willingness to pay for the ecological improvement schemes. The study presents policy suggestions for addressing the challenges to ecology and biodiversity due to the establishment of *Eucalyptus* monoculture plantations. It also provides rational strategies to policy makers for the possibility of ensuring the ecological sustainability and biodiversity conservation in the study area. In contrast to studies including [32,33,35], which mainly utilized CE for the conservation of conserved areas, the current study faced challenges in designing hypothetical attributes and conceptualizing them regarding restricting the expansion of *Eucalyptus*. However, using the same approach as for the restriction of such invasive species was taken as a good opportunity and exception for this study. Good awareness of most of the respondents to the negative impact of *Eucalyptus* is another opportunity. The problem, however, demands continuous policy efforts for ensuring ecological wellbeing and biodiversity conservation.

## 2. Materials and Methods

### 2.1. Study Location

This study was carried out in the *Eucalyptus* hotspot highlands of northwestern Ethiopia (Figure 1). Geographically, it extends from 10°21′23.59″–10°37′28.05″ N, 37°40′25.95″–37°53′09.02″ E. The area is characterized by diverse agro-ecology with an altitude ranging from 2500 to 3900 m asl. Generally, a cool humid and sub-humid climatic zonation characterizes the area with unimodal rainfall, average annual rainfall ranging from 1000–1700 mm, and an average annual temperature of 18 °C. The area had a total population of 2.45 million in 2014 with an average population density of 89 persons per square kilometer. The vast majority of the population in the zone (86.5%) lives in rural areas where agriculture is the predominant economic activity [38].

In terms of ecological setup, the area is characterized by distinct landscape features with slopes ranging from nearly flat to very steep (greater than 45%). Most of the land is used for the cultivation of food crops, while monoculture *Eucalyptus* plantations make up the next highest share. Commonly grown food crops in the area include maize, teff (*Eragrostis tef*), wheat, potato, barley, and beans. *Eucalyptus globulus* is the widely planted species for monoculture plantations in the study area [39].

### 2.2. Choice Modeling

Choice modeling is a popularly applied technique in various fields for estimating the passive use value of environmental goods. It is a random utility model that can be used to explore the marginal willingness to pay for all the attributes and levels [40]. The technique was originally developed from conjoint analysis and differs from it in that instead of rating or ranking, it asks respondents to choose one alternative from each of the alternative choice sets, and in this way, it differs from contingent valuation, which focuses on valuation of a specific trade-off.

McFadden [41] defined different choices in a situation as alternative choices, or simply alternatives and every choice is made from a set of alternatives. The environment of the decision maker determines the universal set of alternatives, but single decision makers do not consider all alternatives. In a choice experiment (CE) method, respondents are presented with a series of alternatives, differing in terms of attributes and levels, and are asked to choose the most preferred one. A baseline alternative, corresponding to the “status quo” or “do nothing” situation, is usually included in each choice set. This is because one of the options must always be in the respondent’s currently feasible choice set to be able to interpret the results in standard welfare economic terms.

A CE has many advantages over the contingent valuation method. It is easier to estimate the value of the individual attributes that make up an environmental good. In the contingent valuation method (CVM), the value of individual characteristics of the good will not be estimated and thus estimation is made for the environmental good or services as a whole [42,43]. A CE provides the opportunity to identify the marginal values of attributes, which may be difficult to identify using revealed preference data because of co-linearity or lack of variation. Because of this, a CE offers advantages over the CVM in terms of benefits transfer if environmental goods can indeed be decomposed into measurable attributes with money values and if socioeconomic variables are included in the CE models.

A CE also avoids the limited choice problem of the dichotomous choice design in the CVM, as respondents are not faced with the stark “all or nothing” choice. They may choose one of two environmental alternatives. Thus, in a CE design, there are repeated opportunities for them to express their environmental preferences.

The use of the choice experiment method is a model of consumer choice following the works of Lancaster [44] and the econometric model is basically derived from the random utility theory [41]. The governing idea of the model is that consumers derive satisfaction, not from the goods themselves, but rather from the attributes of those goods. According to this theory, consumption decisions are determined by the utility that is derived from the attributes of a good, rather than from the good in isolation. The econometric ground of the CE pivots on the behavioral framework of random utility theory, which describes discrete choices in a utility-maximizing framework.

### 2.3. The Theory of Choice

According to this theory, individuals are assumed to make choices based on the attributes of the alternatives with some degree of randomness [44]. Random Utility Theory (RUT) states that utility derived by individuals from their choice is not directly observable, but an indirect determination of preferences is possible and thus the utility can be best explained in terms of those attributes. It is also understood that choice is not a static action but has random elements within it [45]. With this arena, the utility function can be decomposed into an observed/measurable component and an unobserved/random component. This model currently serves as the foundation for modeling the choices that individuals make [33,46]. The random utility model allows for random (error) influences in addition to identify fixed ones [41]: (1)$$U_{ij}=V_{ij}+\varepsilon_{ij}$$
where, $u_{ij}$ represents utility derived for consumer *j* from option *i*, $V_{ij}$ is an attribute vector representing the observable component of utility from option *i* for consumer *j*$\varepsilon_{ij}$ is the unobservable component of latent utility derived for consumer *j* from option *i* [36].

In CE, where the respondent is asked to choose the most preferred among a set of alternatives, random utility theory can be used to model the choices as a function of attributes and attribute levels. The RUT thus, provides a link between the deterministic model outlined above and a statistical model [41].

Assuming a linear additive form for the multidimensional deterministic attribute vector ($V_{ij}$):(2)$$V_{ij}=\beta_{1i}f_{1}(S_{1ij})+\ldots +\beta_{ki}f_{k}(S_{kij})$$
where $\beta_{ki}$ are utility parameters for option *i* and $S_{ij}$ represent 1 to *k* different attributes with different levels. Then, by expanding Equation (1), we find:(3)$$V_{ij}=\beta_{1i}f_{1}(S_{1ij})+\ldots +\beta_{ki}f_{k}(S_{kij})+\varepsilon_{ij}$$

This random utility model is converted into a choice model with the assumption that an individual *j* will select alternative *i* if and only if $u_{ij}$ is greater than the utility derived from any other alternative in the choice set. Alternative *i* is preferred to *j* if $$P[(V_{ij}+\varepsilon_{ij\;})>(v_{iq}+\varepsilon_{iq})]$$ and choice can be predicted by estimating the probability of individual *j* ranking alternative *i* higher than any other alternative *j* in the set of choices available [36,47].

The probability of consumer *j* choosing option *i* from a choice set may be estimated by means of the maximum likelihood estimation whereby estimates are obtained through the maximization of a probabilistic function with respect to the parameters [33,36,47].

Then, $V_{ij}$ which is the systematic component, could be specified as a function of the vectors of the restriction strategy attributes *Z* which characterizes *j* that alternative and respondent *i*’s characteristics. As this part is random, individual choices cannot be predicted certainly and this leads to the expression of the probability of choice as:(4)$$P\left(i\right)=P\left(V_{ij}+e_{ij}>V_{im}+e_{im}\right);\forall_{m}\in C\;\;$$

Presuming that the random terms are distributed independently and identically and follow a Gumbell distribution, the probability that alternative h will be selected is estimated with multinomial logit model (MNL) [48] as follows:(5)$$P(i)=\frac{\exp(V_{ij})}{\sum_{j\in C}\exp(V_{ij})}$$

To improve the fitness of the model, violation of irrelevant alternatives need to be avoided using different techniques. In this study, we analyzed the socioeconomic alternatives using a multinomial logit model including socioeconomic attributes with minimum bias and more accuracy by employing the STATA14 program.

In this model, coefficients estimated can be used for estimating the rate at which respondents are willing to choose one attribute for another. The trade-off estimated is known as the marginal willingness to pay or part-worth or an implicit price. This expresses the amounts of money respondents are willing to pay to receive more of the non-marketed environmental attribute [49].
(6)$$Marginal\;Willingness\;to\;Pay=-\frac{\beta_{Non-marketed\;Output}}{\beta_{Monetary\;attribute}}$$

To analyze welfare, the willingness to pay WTP of farmers for a change in attribute levels was estimated by taking the ratio between the coefficients of individual attributes and the price attribute [50].
(7)$$WTP_{i}=\frac{dx_{i}}{dx_{c}}=-\frac{\beta_{i}}{\delta_{c}}$$
where, $WTP_{i}$ = Willingness to pay for a given attribute, $\beta_{i}$ = Marginal utility of an attribute *i*, and $\delta_{c}$ = Estimated parameter of costs associated to the alternatives.

### 2.4. Design of the Survey Questionnaire

Identification of attributes and their levels is the first and essential step in designing a questionnaire for a choice experiment. In this paper, management scenarios for restricting *Eucalyptus* expansion with their attributes were determined in consultation with respondents in the study area. The identification of an appropriate experimental design is the most important precondition for undertaking a CE analysis. Using the right experimental design, it is possible to create choice sets in the most efficient way that combines attribute levels to alternatives and alternatives to the choice sets [42]. Before administering the choice experiment questionnaire, a focus group discussion and pilot survey of 35 respondents was undertaken to agree on the final version of the questionnaire and attributes and levels. Table 1 shows the final attributes and levels for the choice model.

Our assumption was that conservation plans that restrict the expansion of *Eucalyptus* over crop lands would create positive environmental impacts, which were used as attributes of the choice experiment. Complete factorial designs allow the estimation of the full effects of the attributes upon choices that include the effects of each of the individual attributes presented (main effects) and the extent to which behavior relates to variations in the combination of different attributes offered (interactions). As these designs often give an impractically large number of combinations to be evaluated, we reduced the number of scenarios of combinations to an optimum manageable size of nine different choice sets using orthogonality analysis using SPSS software (IBM Corporation, Armonk, NY, USA) [47]. To make the questionnaires manageable for respondents, we further limited these choice sets to three blocks.

With the three attributes, each with three labels, we applied an experimental design technique of 3^3^ combinations for a total of 81 alternatives with main effects, which generated nine orthogonal combinations, which were blocked into three different questionnaire versions, each including three choice sets [51].

### 2.5. Data Sources and Sampling

The sample design strategy entails four distinct steps: selecting the target (sample) population, determining who to sample (the sample frame), determining the appropriate sample size, and choosing the method of respondent selection and elicitation of response technique. Relevant stakeholders were identified through a brainstorming session with respondents with the help of local experts following stakeholder analysis [47].

We considered a final sample of 388 respondents for the final data collection. Then, before administering the choice modeling questions on the respondents, we induced informative discussion about the current situation of *Eucalyptus* expansion and its impact on biodiversity together with elicitation of response techniques [36,52].

Each respondent was provided with three choice sets and left to choose among three alternative scenarios, which showed various options for the restriction of areas of *Eucalyptus* plantation over crop lands and biodiversity conservation strategies in the study area (Figure 2). The alternatives involved the options from minimal to higher conservation strategies with inclusion of payment as annual utility fee.

The payment vehicle was used as a voluntary continuous donation contributed through a yearly land use fee for 3 years, which could catch the present value of preferences for *Eucalyptus* restriction for ecological wellbeing. The payment levels of USD 31.21, 62.41 and 93.62 were determined based on the focus group and pilot survey. Non-attribute variables, which were supposed to have high predictor capacity, were also prioritized for their inclusion in the multinomial logit model (Table 2).

## 3. Results

### 3.1. Characteristics of Respondent Residents

The demographic and socioeconomic characteristics of the respondent households are presented in Table 3. For head of the household, about 90% of sample respondents were male and only 10% were female. Regarding marital status, 80.9% of residents were married, 5% not married, 4% divorced and 3% were widowed. Age is an important demographic factor that determines household involvement in different activities. Considering their age, most respondents (90%) were found to be within the age range of 31 to 65 years, about 5% were under 31 years, and only 5% were above 65 years. On average, families had about six individuals, which was relatively higher than the national average family size. Likewise, 8.93% of respondents could not read and write, 80.51% could read and write, 4% followed their education in religious schools, 5% were grade 1 to 6, and only 1% followed formal schooling of grade 7 to 12.

All farm households involved in this study possessed varying sizes of plots. The average land holding of respondent households was 1.37 ha, which was utilized for the cultivation of food crops, *Eucalyptus* plantations and homestead plantations. We found that farming practices of residents were not limited only to their own plots but they also cultivated a sizable area of land for production of food crops and plantations of *Eucalyptus* on a rental basis. Considering the annual income of residents from different sources, income from *Eucalyptus* was found to be the highest (USD 881.20), accounting for about 46% of their total annual income. Similarly, the average annual income obtained from cultivation of food crops was USD 502.98. Income from non-farm activities (off-farm and remittance) and livestock rearing were found to be UDSD 330.07 and USD 211.43, respectively.

### 3.2. The Multinomial Logit Model

A multinomial logit model was used to analyze the preferences of respondents and their WTP for the ecological improvement schemes (Table 4). The model fulfilled the test of goodness of fit (Prob > Chi^2^ = 0.0000) criteria implying that the predictor attributes under consideration had strong determination and explanatory power. In Models 2 and 3, we introduced socioeconomic variables to the indirect utility function and observed that presence variable as additive forms and interaction with selected variables respectively.

The first model presented in Table 4 represents the basic attribute variables that were specified for the ecological improvement program. In this model, all coefficients were found to be significantly (*p* < 0.01) different from zero. The signs of attribute variables were found as expected. Accordingly, the land use plan, to use a small portion of their plots for plantation increased the probability that an option would be chosen by 42%. Similarly, all the rest of the attribute variables increased the probability. Most of the predicted variables significantly (*p* < 0.01) and positively influenced the preferences of residents for each improvement scheme. In Model 1, the highest (48.95%) probability increase in choice of options was brought by the program for increasing the number of other tree species to be planted.

In Model 2, all the attribute variables and three of the non-attribute variables were found to be significantly (*p* < 0.001) different from zero. Variables including sex (0 = female; 1 = male), land size and the number of oxen owned were found to influence respondent choice negatively, showing that a unit increase in the respective variables causes a decrease in the probability of households’ preferences for the improvement program. For this model, the highest (about 49%) increase in the probability that an improvement option would be chosen was observed for the number of other tree species grown. For the non-attribute variables, the number of family laborers had a positive and significant (*p* < 0.01) effect on respondents’ preferences for improved ecological wellbeing. Results in this model also showed that households with higher income, older in age and with higher number of family laborers had an increased option choice.

Results of Model 3 presented how socioeconomic variables that were significant in Model 2 affected option choice in interactions. In this model, attribute variables were found to significantly (*p* < 0.001 and *p* < 0.05) influence respondents’ choice for the improvement program. Among the attribute variables, the number of other tree species to be planted increased respondents’ choice by 50.13%. The interaction of variables that were significant in Model 2 with attribute variables also showed significant influence on respondents’ preferences except for the interactions of income *land fertility, age *number of other tree species to be grown and family labor *number of other tree species to be grown.

To obtain a better representation and understanding of the effect of attribute and non-attribute variables beyond the individual effects, we also examined their interaction effects on residents’ preferences. Signs of the coefficients of non-attribute variables including sex, size of holdings and the number of oxen owned were found to be negative, but these were non-significant influences on residents’ choice for the ecological improvement program.

### 3.3. Estimation of Marginal WTP Values

We calculated the implicit value of marginal attribute changes by observing the marginal rate of substitution between the price attribute and the respective attribute, i.e., by taking the ratio of the attribute under consideration to the payment coefficient. The marginal willingness to pay (MWTP) measure the amount of money respondents are willing to pay to trade off for a unit improvement in an environmental attribute, or the amount they are willing to pay to prevent the expected welfare losses. For the improvement of the ecological scheme in the study area, the MWTP values represent a change in size of land allocated for *Eucalyptus* (75 to 25% or less), planting *Eucalyptus* on fertile plots to planting on marginal lands, and a change from planting small numbers of other tree species to more diverse tree species for ecological conservation in the study area.

Accordingly, the results showed that residents were willing to trade off a USD 54.83 increase in household annual utility cost for the contribution of the land use plan attribute for the ecological improvement programs.

As seen in Table 5, the compensation variation for improving households’ intended plan of planting *Eucalyptus* on 75% of their plots to 25% was USD 54.83/person/year. Similarly, to change the households’ need for planting *Eucalyptus* on fertile plots to planting on more marginal ones, the corresponding compensation variation was USD 24.35/person/year. The compensation variation for enhancing the number of other tree species to be grown, which is the highest value of all, was USD 125.82/person/year.

This study estimated the amount residents of the study area can willingly pay for a policy relevant to ecological improvement schemes. Overall, residents were WTP USD 205.00/person/year on average as a yearly utility fee in support of the ecological improvement schemes in the study area. The WTP estimates for the ecological improvement programs suggest that residents of the study area were diverse in terms of their views regarding ecological wellbeing.

## 4. Discussion

The ecosystem in the *Eucalyptus* hotspots of the highlands of northwestern Ethiopia is influenced by intensive human activities, land cover changes, and subsequent climate change and loss of biodiversity. This situation will get worse and may result in ecological deterioration unless certain improvements and conservation schemes are not enforced. Recently, rapid land cover changes have been observed in the study area, where *Eucalyptus* was alarmingly replacing arable land in monoculture plantations, which has impacted the ecosystem and brought transformation of the cultural landscapes in the area.

Though the majority of the land in the study area is primarily used for the cultivation of food crops, livestock rearing and plantations, it also provides services including biofuel, fiber, opportunities to store carbon, biodiversity, and recreational and aesthetic values [53,54]. Nevertheless, monoculture plantations of *Eucalyptus* can result in disturbance of the natural ecosystem and loss of ecosystem services [55]. This infers that the conservation of biodiversity and ecosystems is quite important for ensuring the improved livelihood and social welfare of the population [56].

In the study area, the area in which *Eucalyptus* is grown is expanding rapidly. Studies by Rejmanek and Richardson [57] showed that *Eucalyptus* has reached an estimated coverage of about 2.3 million hectares globally since its introduction from its center of origin. In Portugal, for instance, where the species seems to find particularly favorable conditions for successful reproduction, an estimated area of 845,000 ha, which accounts about 26% of the Portuguese forest, is covered with *Eucalyptus* plantations [58]. Despite its spread worldwide, *Eucalyptus* has proven to be particularly successful in tropical and sub-tropical regions like Portugal and Spain, more than elsewhere [59]. Because of suitable climatic conditions, *Eucalyptus* grows well and is well-adapted to conditions in Africa, including Ethiopia.

The replacement of crop lands by *Eucalyptus* can also apparently alter the ecological make-up of the area by changing the scene and population dynamics of flora and fauna. As a non-native species, the expansion of *Eucalyptus* can result in the transformation of the landscape and a change in the natural ecosystem. This finding is also in line with Toledo et al. [60], who assessed the potential spread and invasive nature of *Eucalyptus*, which could impact ecosystem properties and functions

To limit the expansion of *Eucalyptus* and improve the disturbed natural ecosystem in the study area, endorsing ecological improvement schemes towards sustainability of the ecosystem and biodiversity would be essential. Residents of the study area are aware of that *Eucalyptus*, as a non-native tree species, has a potentially invasive behavior. This conclusion is in line with the study by Vance et al. [61], which showed the negative impacts of the tree on biodiversity can lead to long-term transformation of cultural landscape of an area. Large-scale *Eucalyptus* plantations have caused various problems including reduced species diversity and loss of soil nutrients, which threaten ecological security regionally and worldwide [62,63]. In addition, it would also be rather necessary to estimate the non-market benefits of the ecosystem to ensure sustainable functioning of the agricultural and non-agricultural landscape though examining residents’ preferences and attitudes to the conservation of the ecosystem and biodiversity. However, residents’ preferences and WTP for the ecological improvement schemes have not been studied. To this end, the development of attributes that can help improve the ecological wellbeing would be important.

Results presented in this study showed that residents of the *Eucalyptus* hotspot highlands of northwestern Ethiopia had diverse preferences to ecological improvement and WTP for the same programs. Multinomial logit results indicated that all attribute variables significantly and positively influenced residents’ willingness to pay for the improvement programs. This conclusion is in line with the studies by [64,65,66] that reported significant variations in preferences of residents and tourists regarding the Green Island environmental resources in Taiwan.

Changes in personal attitudes of residents towards accepting proposed ecological improvement schemes are quite important for sustainable biodiversity conservation. Residents considered in this study differed in terms of demographic variables (such as age, sex, and family labor) and socioeconomic variables (like size of land holdings and average annual income). The changes in residents’ preferences for the ecological improvement programs were attributed to these variations. However, there was limited information regarding residents’ preferences and their WTP for the acceptance of ecological improvement schemes focused towards improving the ecological and biodiversity wellbeing.

Results of separate model runs showed different effects of attribute and non-attribute variables. The significance of all attribute variables in the first and second run showed that the explanatory powers of the attributes were as expected. Among the socioeconomic variables that were included in the second model, income source, age, and number of family laborers were found to significantly influence residents’ choice. The findings also suggest that residents who better made choices of improved ecological conservation were those who had more income sources, were older in age and had more family laborers. This finding is consistent with the study by [67] who reported that income level and age of respondents had significant impact on WTP for the conservation of the natural heritage in Tatra National Park. Sex, size of land holding, and the number of oxen owned appeared with negative coefficients, implying that residents owning a greater number of oxen involved in cultivation of food crops preferred assigning a larger share of their plots to cultivation of food crops, than to plantation of *Eucalyptus*.

For all the improvement schemes, residents showed varying levels of preferences. The schemes were designed for improving the level of ecological security and were comprised of three attributes and their levels including reduction of total area of *Eucalyptus* plantations, excluding for *Eucalyptus* plantations, and increasing the number of other tree species to be grown.

We estimated the MWTP of respondents by taking the implicit value of marginal attribute changes with respect to the payment coefficient [68]. It measures the amount of money respondents are willing to pay to trade off for a unit improvement in an environmental attribute, or the amount they are willing to pay to prevent the expected welfare losses. Residents were WTP on average more for increasing other tree species (third attribute) and were WTP less for the attribute encouraging plantation of *Eucalyptus* on less fertile lands for the improvement program (second attribute). Accordingly, there was a 26.75% change in MWTP of respondents for the first attribute (i.e., intended land use plan or scale of *Eucalyptus* plantation, from 75–25% or less). The highest attribute change (61.38%) was observed for the third attribute (involving increasing plantation of other tree species from minimal to more). The findings are in line with a study by Kim et al. [69] which assessed the non-market environmental values of biodiversity conservation in Vietnam and indicated that residents were WTP a monthly payment of VND 913 for a % increase in healthy vegetation.

We found that the contribution of a reduction of plantation area of *Eucalyptus* was estimated to be USD 54.83/person/year. Similarly, the benefits from the exclusion of fertile plots from plantations and increasing the number of other tree species were USD 24.35/person/year and USD 125.82/person/year, respectively; the overall value being USD 205. Differences in attribute values have also been reported by [67].

We found that respondents’ preference for the ecological improvement programs can be improved if land allocated for *Eucalyptus* was significantly reduced, fertile lands were set aside from *Eucalyptus* plantation and the number of other tree species was increased.

Our study showed that the socioeconomic backgrounds of the respondents contributed to the heterogeneity in residents’ preferences and their WTP for ecological improvement schemes. This outcome is in line with the study by [70]. Our findings are similar to the results of the study by [71], which used a similar approach to evaluate the attitudes and WTP values of local residents for deep-sea ecosystem conservation in England.

This study is a first attempt to evaluate the economic value of restricting areas of monoculture *Eucalyptus* plantation. Our findings can be utilized for instructing residents and the provision of information regarding conservation plans for ecosystems and biodiversity.

Nonetheless, we acknowledge the uncertainties with our approach. For instance, our study participants were confronted with a hypothetical bias in which there was no real transactions and thus there might be over estimations.

## 5. Conclusions

This study presents economic estimation of restricting the area of plantations of *Eucalyptus* in the *Eucalyptus* hotspot highlands of northwestern Ethiopia from the perspective of ecological economics using a CE method and devised attributes that are important for ecological improvement and biodiversity conservation. The study indicated that policy-targeting management advice is essential for the development of a sustainable ecosystem, especially in areas that are fragile in terms of hosting biotic and abiotic components that are essential to healthy functioning of the ecosystem. The findings of this study revealed that the expansion of areas of monoculture *Eucalyptus* plantation is a big threat for ecological wellbeing and biodiversity in the area. A study by Foresmaire [16] outlined the invasive nature of *Eucalyptus* in Portugal and part of Spain.

In this study, a choice experiment approach was validated for constructing a random utility model and examining the preferences of residents to possible ecological improvement options. The proposed attribute and non-attribute variables were identified by considering previous works including those by Kragt and Bennett, [35] and Luisetti [37] and the views of residents who participated in the ecological improvement program. It is believed that this study presents a splendid prospect for employing a choice modeling approach for addressing real policy-impacting environmental challenges involving passive use values and a plausible payment vehicle. The results showed high heterogeneity in preference and WTP of residents for the proposed ecological improvement schemes. Luisetti et al. [37] drew similar conclusions while examining the readjustment of coastal policies in England using a CE approach.

The ecosystem of the *Eucalyptus* hotspot highlands of northwestern Ethiopia is highly disturbed due to the undergoing dramatic land use changes due to human activity. The study is in line with the finding of Townhill et al. [29] who reported the burden on the Arctic species and the ecosystem resulting from human activity. Absence of land policy initiatives that work for proper land use and limited management options have contributed to the unprecedented expansion of areas of monoculture *Eucalyptus* plantations and ecological deterioration. Validation of the CE approach showed that respondents are willing to pay for the proposed ecological improvement programs.

We believe that this study yields important findings that can provide trustworthy information to policymakers and researchers and creates public awareness regarding ecological wellbeing and biodiversity conservation. Our findings can address similar areas that demand immediate policy interventions regarding management of development strategies for a given ecosystem and biodiversity.

The findings can provide important implications about relevant attributes for ecological improvement. Based on our findings, the following key policy suggestions are forwarded for the ecologically sensitive *Eucalyptus* hotspot areas of the highlands of northwestern Ethiopia. (1) Policies should restrict areas of further plantations of *Eucalyptus*. (2) Emphasis should be given to land types for the plantation of *Eucalyptus* setting aside cultivable lands. (3) Focusing on increasing plantation areas of other tree species. (4) Creating awareness on residents regarding the conservation of the ecosystem through restricted plantation areas of *Eucalyptus* and increasing plantation areas of other tree species. Finally, it is suggested that further research shall be carried out as extension of this study.

## Figures and Tables

**Figure 1 ijerph-19-09073-f001:**
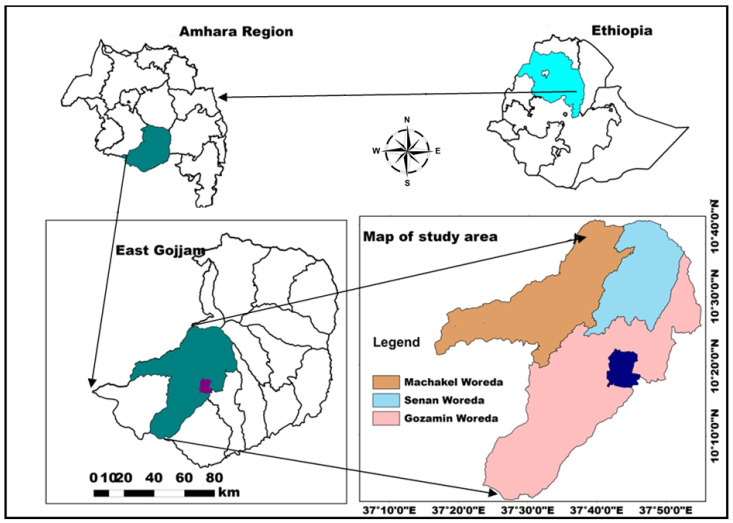
Map of the study area.

**Figure 2 ijerph-19-09073-f002:**
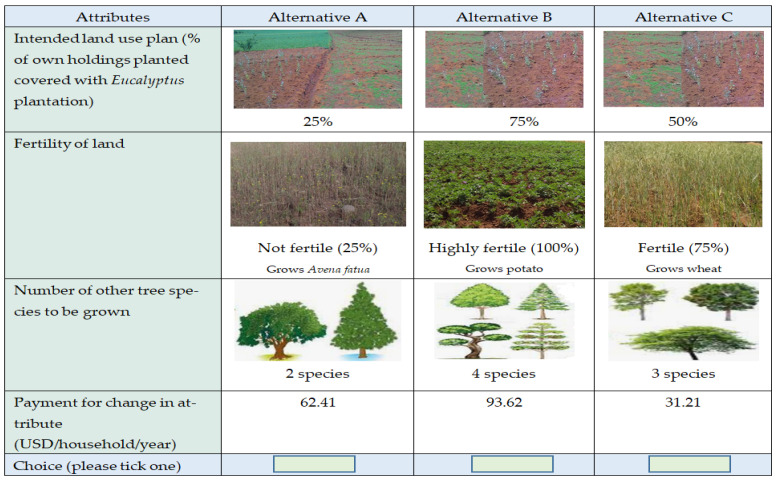
Sample profile of choice sets.

**Table 1 ijerph-19-09073-t001:** Conceptual attributes and levels.

Attribute	Description/Assumptions	Levels
Intended land use plan (scale of *Eucalyptus* plantation, %)	Plots devoid of *Eucalyptus* plantations are rich in biodiversity	25
		50
		75
Fertility of land ^1^	Plantations focusing mainly on marginal lands will have reduced impact ecosystem	Not fertile
		Fertile
		Highly fertile
Number of other tree species to be grown	The more the number of other tree species, the better will be the ecological wellbeing	2
		3
		4
Payment for change in attribute (USD)	How much are households willing to pay as a compensation for restriction of *Eucalyptus* plantation	31.21
		62.41
		93.62

The ‘numeraire’ used throughout this paper are in USD using the 21 January 2020 exchange rate ($1 USD = ETB 32.044. ^1^ Land fertility is a relative term, which is based on farmers’ ratings. Marginal lands that are not used for the cultivation of food crops, except avena and lupin, are categorized as “*not fertile*”. Lands that support the cultivation of wheat, barley, flax, etc. are categorized as “*fertile*”. “*Highly fertile*” lands exceptionally permit the cultivation of maize, potato, bean and teff.

**Table 2 ijerph-19-09073-t002:** Summary of variables used in the choice model.

Variable	Description
Attribute variables	
Intended land use plan (scale of *Eucalyptus* plantation, %)	Plots devoid of *Eucalyptus* plantations are rich in biodiversity
Fertility of land planned for *Eucalyptus* plantation (%)	Plantations of *Eucalyptus* on marginal lands will have reduced impact on biodiversity
Number of other tree species to be grown	The more the number of other tree species, the better will be the ecological wellbeing
Payment (Birr)	Charge incurred as a compensation for restriction of *Eucalyptus* expansion per hectare year
Non-attribute variable	
Income source	Sources of income from crop, livestock, trading, rentals, etc.
Age	Age of household head
Sex	Sex of the household head
Family Labor	Number of family labor available (continuous)
Total land size (ha)	Size of total land holdings of households (ha)
Slope (%)	Slope of cultivable plots (%)
Number of oxen	Number of oxen owned by a household

**Table 3 ijerph-19-09073-t003:** Sociodemographic characteristics of respondents.

Characteristics	Variable	Value
Demography	Average family size	6
	Dependency ratio (%)	64
	Age	48
	Gender (%)	
	Female	10
	Male	90
	Marital status (%)	
	Married	80.9
	Not Married	5.2
	Divorced	7.7
	Widowed	6.2
Literacy status (%)	Cannot read and write	8.93
	Read and write	80.51
	Religious school	4.12
	Grade 1–6	5.15
	Grade 7–12	1.29
	Average land size (ha)	1.37
Farm resource and income	Average annual income (USD) from	
	Crop	502.98
	*Eucalyptus*	881.20
	Livestock	330.07
	Non-farm sources	211.43

**Table 4 ijerph-19-09073-t004:** Result of the multinomial logit model.

Attribute Variables	Model 1	Model 2	Model 3
	Coefficient	Coefficient	Coefficient
Land use plan	0.42199 (0.05109) ***	0. 42128 (0.05139) ***	0.41006 (0.05142) ***
Fertility	0.23473 (0.05459) ***	0.23437 (0.05493) ***	0.22117 (0.06120) **
Other trees	0.48953 (0.05362) ***	0.49173 (0.05393) ***	0.50125 (0.05421) ***
Payment	0.00031 (0.00003) ***	0.00031 (0.00003) ***	0.00033 (0.00004) **
Constant	−3.32501 (0.19608) ***		
Income		0.10495 (0.04776) ***	0.12136 (0.04991) ***
Age		0.01499 (0.00498) ***	0.01501 (0.00611)
Sex		−0.15990 (0.13622)	−0.16030 (0.14501)
Family labor		0.16043 (0.03937) ***	0.16171 (0.04015) ***
Land size (ha)		−0.00366 (0.06307)	−0.00386 (0.08821)
Slope (%)		0.11086 (0.08254)	0.11104 (0.08715)
Number of oxen		−0.11045 (0.05235)	−0.11630 (0.06104)
Constant		−4.90173 (0.41072) ***	
Income *Land use plan			0.31090 (0.03281) **
Income *Fertility			0.03931 (0.01410)
Income *Other trees			0.21095 (0.09142) ***
Age *Land use plan			0.02657 (0.03120) ***
Age *Land fertility			−0.03531 (0.02381) **
Age *Other trees			0.31140 (0.00201)
Family labor *Land use			0.07142 (0.03224) ***
Family labor *Fertility			0.21705 (0.03496) **
Family labor *Other trees			0.362510 (0.04184)
Constant			−6.03272 (0.49035) ***
Log likelihood	−3406.2996	−3382.7425	−3391.5241
Pseudo R^2^	0.0458	0.0528	0.0762

Number of obs = 3492; Prob > Chi^2^ = 0.0000; Parenthesized figures are standard deviations; ** and *** indicate level of significance at 5% and 1% probability. * in between two variables indicate interaction.

**Table 5 ijerph-19-09073-t005:** MWTP for a change in attributes.

Change in Attribute	MWTP (USD/Person/Year)
Intended land use plan (scale of *Eucalyptus* plantation, from 75–25% or less)	54.83 (26.75%)
Shift of plantation from fertile to non-fertile plots	24.35 (11.88%)
Increasing plantation of other tree species from minimal to more	125.82 (61.38%)
Total	205.00 (100%)

Note: MWTP values are converted to USD after running the multinomial logit model.

## Data Availability

The data supporting the results presented in this study can be obtained from the corresponding author on request.

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
