# Peer review of "Estimating the Economic Values of Restricted Monoculture Eucalyptus Plantations: A Choice Modeling Approach"

_ijerph, 2022, doi:10.3390/ijerph19159073_

Round 1
Reviewer 1 Report
The manuscript is of interest; it is well constructed and written. In my opinion it deserves publication in its present form. I only suggest using Eucalyptus in italics troughout the work.
Author Response
Dear anonymous reviewer,
Thank you for your kind feedback and suggestion to our manuscript.
We have considered your comment and corrected the manuscript accordingly.
Kind regards,

Reviewer 2 Report
The authors examined the preferences of residents of the Northwestern Ethiopia and estimated their willingness to pay (WTP) for the proposed ecosystem improvement programs using a Choice Experiment (CE) approach. This manuscript is well organized and the drawn conclusions are coherent with the obtained results. This study can provide vital policy implications and contribute to knowledge as it presents how the non-market valuations of ecosystems help maximize social welfare.
Line 22: To delete (CE)…you don’t use it another time in the abstract.
Line 28: To delete (MWTP)…you don’t use it another time in the abstract.
Lines 93 – 95: I think that you should add these recent references to support this your sentence: “Ongoing land degradation, climate change, ocean acidification, and land cover changes are reducing the quantity and quality of ecosystem services being provided globally”. I would like to suggest:
Ali, H., et al. (2021). Expanding or shrinking? Range shifts in wild ungulates under climate change in Pamir-Karakoram mountains, Pakistan. PloS one, 16(12), e0260031.
Townhill, B. L., et al. (2021). Pollution in the Arctic Ocean: An overview of multiple pressures and implications for ecosystem services. Ambio, 1-13.
Lines 124 – 131: Please, explain better your hypothesis and predictions.
Lines 135 – 148: To add the geographical coordinates of your study area.
Line 151: It is study area not study are.
Lines 152 – 252: You can move this part of the manuscript in the supplementary materials.
Lines 409 – 498: You should discuss your results comparing them also with other study focused on other species in other geographic areas in the world.
Author Response
Dear anonymous reviewer,
Thank you for your scientific comments and suggestion to our manuscript.
We have considered your comments one by one as can be seen in track changes in the manuscript and the attached author response file.
Thank you,
Amare

Reviewer 3 Report
- Your abstract should clearly state the essence of the problem you are addressing, what you did and what you found and recommend. That will help a prospective reader of the abstract to decide if they wish to read the entire article.
- The authors should clearly illustrate the challenges and opportunities of the research direction, compared to other available studies.
- In the method, the questionnaire designed for this study could be placed in the supporting information.
- The suggestion for future research is missing in the conclusion section.
- Avoid using first person in the paper text (e.g. we, our, he, etc..........).
Author Response
Dear anonymous reviewer,
We are thankful for your scientific suggestions and comments for our paper.
We have considered all your comments as can be seen in the attached author response file and in track changes in the manuscript.
Thank you,
Amare

Round 2
Reviewer 3 Report
The presented version of this paper, corrected according to the reviewer's suggestion is now well written, the discussion part was improved and looks much better, and in my opinion, it can be published. I have no more comments on this manuscript and I am convinced that it will be very interesting for international readers.